# Analysis of Dynamic Evolution and Spatial-Temporal Heterogeneity of Carbon Emissions at County Level along “The Belt and Road”—A Case Study of Northwest China

**DOI:** 10.3390/ijerph192013405

**Published:** 2022-10-17

**Authors:** Shaoqi Sun, Yuanli Xie, Yunmei Li, Kansheng Yuan, Lifa Hu

**Affiliations:** 1College of Urban and Environmental Sciences, Northwest University, Xi’an 710127, China; 2Space Planning and Large Data Research Center of One Belt and One Road, Northwest University, Xi’an 710127, China; 3Shaanxi Key Laboratory of Earth Surface System and Environmental Carrying Capacity, Northwest University, Xi’an 710127, China

**Keywords:** carbon emission, energy consumption, nighttime light data, spatial-temporal heterogeneity, two-stage nested Theil index, geographically and temporally weighted regression

## Abstract

Northwest region is the main energy supply and consumption area in China. Scientifically estimating carbon emissions (CE) at the county level and analyzing the spatial-temporal characteristics and influencing factors of CE in a long time series are of great significance for formulating targeted CE reduction plans. In this paper, Landscan data are used to assist NPP-VIIRS-like data to simulate the CE from 2001 to 2019. Spatial-temporal heterogeneity of CE was analyzed by using a two-stage nested Theil index and geographically and temporally weighted regression model (GTWR). The CE in northwest China at the county increases yearly while the growth rate slows down from 2001 to 2019. The spatial pattern forms a circle expansion centered on the high-value areas represented by the provincial capital, which is also obvious at the border between Shaanxi and Ningxia. Axial expansion along the Hexi Corridor is conspicuous. The spatial pattern of CE conforms to the Pareto principle; the spatial correlation of CE in northwest counties is increasing year by year, and the high-high agglomeration areas are expanding continuously. It is an obvious high carbon spillover effect. Restricted by the ecological environment, the southwest of Qinghai and the Qinling-Daba Mountain area are stable low-low agglomeration areas. The spatial pattern of CE in northwest China shows remarkable spatial heterogeneity. The difference within regions is greater than that between regions. The “convergence within groups and divergence between groups” changing trend is obvious. According to the five-year socioeconomic indicators, the economic scale (GDP), population scale (POP), and urbanization level (UR) are the main influencing factors. The direction and intensity of the effect have changed in time and space. The same factor shows different action intensities in different regions.

## 1. Introduction

The surge of CE has triggered global climate and environmental change, posing a serious threat to the development of human economic society. Reducing carbon emissions and developing a green, low-carbon economy has become a global consensus [1,2]. As the largest developing country all over the world, China’s rapid urbanization and industrialization have led to long-term high growth in energy consumption [3]. The data released by the Global Carbon Accounting Database indicate that China’s CE in 2020 was 10.67 billion tons, accounting for 30.65% of global CE, which made China one of the major global carbon emitters [4]. China has actively undertaken important responsibilities for ecological environment management and social sustainable development. The “double carbon” target with 2030 and 2060 as the time nodes was formally proposed by China at the 75th session of the UN General Assembly [5]. CO_2_ produced by energy combustion and industrial processes is the main source of global carbon emissions, and the northwest region is the main source of energy supply in China. The gradient economic development mode makes the northwest region take over the high-carbon industries transferred from the middle and east regions for a long time and become the main energy consumption place. There are differences in economic scale, technological level, urbanization process, and resource endowment among counties in northwest China, which lead to differences in carbon emission level, carbon emission intensity, and driving factors among counties. Therefore, it is necessary to explore the spatial differentiation and driving factors of carbon emissions in northwest China at the county scale so as to formulate a more scientific and targeted low-carbon development policy.

The scientific formulation of carbon emission reduction policy depends on reasonable estimation and spatialization of carbon emissions [6]. The IPCC carbon emission coefficient method is widely used in the estimation of carbon emissions in countries and regions because of its simple calculation method and comprehensive consideration of energy consumption in various industries and terminal energy consumption [7,8,9]. This method mainly relies on the source consumption data published by the government. Limited to the “top-down” statistical system of energy consumption in China, most research is focused on national and provincial scales. With the development of remote sensing technology, nighttime light data are widely used in carbon emission fitting research because of its high spatial correlation with carbon emissions [10,11,12,13]. The fitting of carbon emissions by nighttime light data can overcome the differences between the statistical caliber and accounting standards of energy data across administrative boundaries in a wide range of research scales [14,15]. However, it can only identify human social and economic activities in light areas [16]. Carbon emissions in dark areas cannot be captured. However, according to relevant research, about 1.6 billion people around the world live in dark areas that cannot be detected by nightlight remote sensing [17]. The number of people without access to electricity in China reached about 546.7 million in 2000 [18]. It was not until 2015 that China completely solved the problem of 2.73 million people without electricity. Northwest China is the region with the slowest economic development and also the main area where the population without electricity is distributed. Previous studies on the estimation of carbon emissions in northwest China only used nighttime light data to build a fitting model, ignoring the carbon emission sources in dark areas. How to reasonably allocate the total carbon emissions to the dark areas has become an important prerequisite for heterogeneity research.

Spatial differentiation is one of the basic characteristics of geographical phenomena. Taking administrative regions as the basic research unit, it is a conventional idea to analyze the spatial-temporal differentiation characteristics of carbon emissions and their driving factors using exploratory spatial data analysis and regression analysis. At the global scale, in order to reduce the error caused by the global analysis, some scholars tried to group countries based on characteristics such as the level of economic development or resource endowment [19,20,21,22] and then studied the differences between the leading factors between the global and the groups. Analyzing the spatial heterogeneity of carbon emissions based on the global model covered the spatial heterogeneity characteristics within the global. The analysis results may be disturbed by the hybrid effects of spatial data or even draw wrong conclusions. Some scholars realized this disadvantage and the impact of the sample scale on the research results, thus transforming the research area from a global and national scale to a smaller provincial, municipal, and even county scale. Pang et al. and Zhang et al. respectively divided China into eight economic zones and three sub-regions to analyze the spatial differences in carbon emissions [23,24] and concluded that China’s carbon emissions had obvious regional differences. In addition, some scholars have found that China’s carbon emissions have obvious spatial aggregation characteristics and spillover effects at the provincial level [25,26]. On the basis of this analysis theory, Moran’s I index, Markov chain, and spatial Durbin model were used to analyze the temporal and spatial evolution characteristics and driving factors of carbon emissions in China [27,28,29,30]. An urban agglomeration is the main area where human activities and production gather, so it is also the hot spot of carbon emission research. Qian et al. [31], Chen et al. [32], and Wang et al. [33] analyzed the heterogeneity characteristics of the influencing factors of carbon emission in an urban agglomeration and put forward scientific control suggestions based on the differentiated analysis results. County-level research is more detailed. Shi et al. think that the urbanization rate and population management at the county level can better control carbon emissions [34]. According to Wang et al. [35] and Wang et al. [36], a county is an important unit for formulating differentiated and targeted carbon emission reduction policies in China. The panel quantile regression and geographic detector were used to analyze the influencing factors of carbon emissions in China county. They attempted to analyze the relationship between socioeconomic attributes and carbon emissions from the global perspective combined with the spatial attributes of geographic data. They provided quantitative analysis results of each driving factor, but they were unable to deeply analyze the action intensity of driving factors in each county unit, and it was difficult to satisfy the formulation of detailed and differentiated emission reduction policies at the county level.

In summary, the above research provides a theoretical basis for carbon governance, but there are also some shortcomings: Due to China’s top-down energy statistics system, the research is often limited to the national and provincial levels, which cannot provide more powerful support for regional and differentiated carbon emission reduction policies; The inversion of carbon emissions by nighttime light data ignores the dark areas with human activities, which leads to the global carbon emissions not being reasonably spatialized; The existing researches at county level mostly adopt the analytical idea based on the global hypothesis, which masks the heterogeneity of individuals. It is difficult for the macro-level research to accurately grasp the micro-level details. In order to make up for the shortcomings of the above research, the main research purposes of this paper are as follows: (1) based on IPCC algorithm, the carbon emissions of the northwest counties in the long time series from 2001 to 2019 are estimated by using nighttime light data and population data; (2) analyze the spatial-temporal evolution law of carbon emissions in northwest China in 2001, 2005, 2010, 2015 and 2019, and construct the two-stage nested Theil index to explore the difference law of carbon emissions in different scales according to the hierarchical structure of “region–province–city–county”; (3) taking the county as the basic research unit, the GTWR model is used to explore the spatial-temporal heterogeneity of carbon emission drivers from two dimensions of time and space, so as to fully grasp the individual characteristics and non-stationarity of carbon emissions in northwest counties, and provide the basis for regional and differentiated carbon emission reduction policies in northwest China.

## 2. Materials and Methods

### 2.1. Study Areas

The northwest region lies in the hinterland of northwest China (Figure 1). The Second Asia-Europe Continental Bridge runs through it, an important trunk line of the “One Belt and One Road”, an important hub connecting China with Central Asia and European countries. Guanzhong-Tianshui economic zone, Yinchuan Plain, Hexi Corridor, and urban agglomeration on the northern slope of the Tianshan Mountains are the economic cores in northwest China. Xinjiang, Qinghai, Gansu, Ningxia, and Shaanxi are the main provinces, which constitute some 31.71% of the total Chinese land area. The region is rich in mineral resources, with 14.57%, 36.19%, and 37.33% of the country’s proven reserves of coal, oil, and natural gas, respectively [37]. It is also China’s main industrial base and the source of the ”West-East Coal Conveyance”, “West-East Gas Transmission”, and “West to East Power Transmission”. In 2018, the energy production in northwest China was about 888.64 million tons, about 24% of the total energy production in China [38]. The westerly wind prevails in northwest China all year round. Due to the influence of atmospheric circulation, a large number of air pollutants and greenhouse gases produced by the energy and chemical industry are transported to the east, which aggravates the atmospheric pollution in the eastern region [39]. The carbon emission control in the northwest is not only related to local, sustainable development but also has important significance for the effectiveness of environmental management in the eastern region.

### 2.2. Materials

Except for the CE published on the CEADs for validation, the other data are available from 2001 to 2019. The data are finally unified at the county level. The data source and detailed description are shown in Table 1. The usage of the data is displayed in the flow chart (Figure 2).

### 2.3. Methods

The overall logic of this paper is as follows (Figure 2), and the details of the methods involved are listed in the following text.

#### 2.3.1. CE Estimation

The study used the CE Factor (CEF) method published by IPCC to calculate the CE from energy consumption. The corresponding CE coefficient and coal conversion factors for each energy source are shown in Table 2. Considering that previous studies only used nighttime light data in estimating CE and ignored the CE in the dark areas of human activities, the study chose the improved model proposed by Liu et al. [18]. to estimate CE using population data and nighttime light data.
(1)Considering the variability of CE per unit DN and per capita in different regions, the study area is divided into urban area (*S_U_*) and rural area by using a dynamic threshold method based on NPP-VIIRS-like data, according to the statistical data of urban built-up areas in China Statistical Yearbook. Then, the rural area is divided into bright rural area (*S_LR_*) and dark rural area (*S_BR_*). *DN_i_* is the night light brightness value of pixel i; *f* (*DN_i_*) is the area where the light brightness is higher than *DN_i_*, and E(*DN_i_*) is the difference between the extracted urban built-up area (*S_U_*) and the urban built-up area in the statistical yearbook (*S_stat_*). Assume that *E*(*DN*) is the minimum, and the corresponding light brightness value is *DN_j_*. *DN_j_* is the best threshold value. The area where the light brightness value is greater than *DN_j_* is the urban built-up area (*S_U_*), and the area where the light brightness is less than *DN_j_* and greater than 0 is the light rural area (*S_LR_*). The other areas are the dark rural area (*S_BR_*).
(1)$$S(DN_{i})=\sum_{j}^{D_{\max}}f(DN_{j})$$
(2)$$E(DN_{i})=S(DN_{i})-S_{stat}$$
(3)$$\left|E(DN_{i-1})\right|>\left|E(DN_{i})\right|<\left|E(DN_{i+1})\right|$$(2)The main energy consumption purposes of urban and rural are different. According to the energy balance table of the five provinces in China’s Energy Statistics Yearbook, the CE of each province is divided into urban CE (*CO_2U_*) and rural CE (*CO_2R_*) based on the type of terminal consumption. Urban energy terminal consumption includes industrial, construction, transportation, warehousing and postal services, wholesale and retail, accommodation, and various types of energy consumption in urban residents’ lives. Rural energy terminal consumption includes agricultural, forestry, animal husbandry, fishery, and various types of energy consumption in rural residents’ lives. The calculation method is based on the IPCC CE coefficient:(4)$$CO_{2U}/CO_{2R}=\frac{44}{12}\sum_{i=1}^{n}K_{i}E_{i}Z_{i}$$
where: *CO_2U_* and *CO_2R_* are CE in urban and rural areas, respectively; *E_i_* is the consumption of energy i, unit: ton (t); *Z_i_* is the coal conversion factor from energy i to standard coal, unit: (t standard coal/t); *K_i_* is the carbon emission coefficient of energy i (t carbon/t standard coal); The consumption units of natural gas and electricity are, respectively, cubic meter (m^3^) and kilowatt hour (kW. h), and the corresponding coal conversion factors are kg/cubic meter (kg/m^3^) and kg/kilowatt hour (kg/kW. h). The coal conversion factors and carbon emission coefficients of various energies are shown in Table 2.(3)The rural area contains dark areas that people inhabit. In order to reasonably allocate CE to this part, it is necessary to further decompose the *CO_2R_* into total CE in light rural areas (*CO_2LR_*) and dark areas (*CO_2BR_*) with the auxiliary of Landscan population data. In this step, the total population of urban areas (*P_U_*), light rural areas (*P_LR_*), and dark rural areas (*P_BR_*) is extracted by the regional types divided in step (1). The decomposition algorithm of CE in each region is as follows:(5)$${CO}_{2LR}=\frac{C_{2R}}{P_{LR}+P_{BR}}\times P_{LR}$$
(6)$${CO}_{2BR}=\frac{C_{2R}}{P_{LR}+P_{BR}}\times P_{BR}$$(4)Gridding. The light area is gridded by the light brightness value, and the dark area is gridded by the population. The CE per pixel in the light area of each province is *CO_2Lg_*, and that of the dark area is *CO_2Bg_*. The sum of light brightness in urban areas and rural areas with lights is *L_U_* and *L_LR_*, respectively. *L_i_* and *P_i_* represent the light brightness value and population of pixel i, respectively. The grid calculation formula is as follows:(7)$${CO}_{2Lg}=\frac{CO_{2U}+CO_{2LR}}{L_{U}+L_{LR}}\times L_{i}$$
(8)$${CO}_{2Bg}=\frac{CO_{2BR}}{P_{BR}}\times P_{i}$$(5)The CE grid is calculated at the county level to generate the CE in the five northwest provinces at county level:(9)$$CO_{2i}=\sum_{i=1}^{n}(CO_{2Lgi}+CO_{2Bgi})$$

#### 2.3.2. Spatial Autocorrelation Analysis

The global Moran’s I index is often used to characterize the overall aggregation effect of spatial elements within a global domain and can measure the average degree of spatial variation between the unit and the whole within the region [31]. The global Moran’s I index can only reflect the spatial dependence of similar attributes in the study area. Lisa’s clustering diagram can visually reveal the spatial correlation of local units.
(10)$$I_{g}=\frac{n\sum_{i=1}^{n}\sum_{j=1}^{n}w_{\mathrm{ij}}\left(x_{i}-\bar{x}\right)\left(x_{j}-\bar{x}\right)}{\left(\sum_{i=1}^{n}\sum_{j=1}^{n}w_{ij}\right)\sum_{i=1}^{n}{\left(x_{i}-\bar{x}\right)}^{2}}$$
(11)$$I_{l}=\frac{n\left(x_{i}-\bar{x}\right)\sum_{j=1}^{n}w_{ij}\left(x_{j}-\bar{x}\right)}{\sum_{i=1}^{n}{\left(x_{i}-\bar{x}\right)}^{2}}$$
where: *I*_g_ is the global Moran’s index, *I*_l_ is the local Moran’s index, n is the number of counties in Northwest; *W_ij_* is the weight from county i to county j; *x**_i_* − $\bar{x}$ and *x**_j_* − $\bar{x}$ respectively represent the deviation between the observed value and the average value of CE in the *i*th and *j*th county.

#### 2.3.3. Two-Stage Nested Theil Index

This paper refers to the three-stage nested Theil index [42], takes the county as the basic research unit, and decomposes the northwest region into four levels of “region–province–city–county”. The spatial differences of CE in the study area are revealed by decomposing the overall Theil index into three levels: between-provincial differences (*BP*), between-city differences (*BS*), and between-county differences (*BC*).

Assume that *i*, *j*, and *k* represent the labels of provinces, cities, and counties, respectively, take county as the basic unit, the traditional Theil index is decomposed as follows. Where: *C* represents CE; *X* can be expressed as population or GDP. When *X* is population, *T* is the Theil index based on per capita CE; If *X* is GDP, *T* is the Theil index based on CE intensity:(12)$$\begin{array}{l} T=\sum_{i}\sum_{j}\sum_{k}\left(\frac{C_{ijk}}{C}\right)\log\left(\frac{C_{ijk}/C}{X_{ijk}/X}\right)=\sum_{i}\sum_{j}\sum_{k}\left(\frac{C_{ijk}}{C}\right)\log\left(\frac{C_{i}/C}{X_{i}/X}\cdot \frac{C_{ijk}/C_{i}}{X_{ijk}/X_{i}}\right)\\ =\sum_{i}\left(\frac{C_{i}}{C}\right)\log\left(\frac{C_{i}/C}{X_{i}/X}\right)+\sum_{i}\sum_{j}\sum_{k}\left(\frac{C_{ijk}}{C}\right)\log\left(\frac{C_{ijk}/C_{i}}{X_{ijk}/X_{i}}\right)=BP+WP \end{array}$$

*BP* is the difference between provinces. *WP* is the difference between counties within the province, and *WP* can be further decomposed:(13)$$\begin{array}{l} \sum_{i}\sum_{j}\sum_{k}\left(\frac{C_{ijk}}{C}\right)\log\left(\frac{C_{ijk}/C_{i}}{X_{ijk}/X_{i}}\right)=\sum_{i}\sum_{j}\sum_{k}\left(\frac{C_{ijk}}{C}\right)\log\left(\frac{C_{ij}/C_{i}}{X_{ij}/X_{i}}\cdot \frac{C_{ijk}/C_{i}{}_{j}}{X_{ijk}/X_{ij}}\right)\\ =\sum_{i}\sum_{j}\left(\frac{C_{ij}}{C}\right)\log\left(\frac{C_{ij}/C_{i}}{X_{ij}/X_{i}}\right)+\sum_{i}\sum_{j}\sum_{k}\left(\frac{C_{ijk}}{C}\right)\log\left(\frac{C_{ijk}/C_{i}{}_{j}}{X_{ijk}/X_{ij}}\right)=BS+BC \end{array}$$

*WP* is decomposed into the difference between cities (*BS*) and the difference between counties differences (*BC*). The overall Theil index is decomposed into the following forms:(14)$$\begin{array}{l} T=\sum_{i}\left(\frac{C_{i}}{C}\right)\log\left(\frac{C_{i}/C}{X_{i}/X}\right)+\sum_{i}\sum_{j}\left(\frac{C_{ij}}{C}\right)\log\left(\frac{C_{ij}/C_{i}}{X_{ij}/X_{i}}\right)+\sum_{i}\sum_{j}\sum_{k}\left(\frac{C_{ijk}}{C}\right)\log\left(\frac{C_{ijk}/C_{i}{}_{j}}{X_{ijk}/X_{ij}}\right)\\ =BP+BS+BC \end{array}$$

#### 2.3.4. Geographically and Temporally Weighted Regression (GTWR)

The GTWR incorporates the heterogeneity relationship of the study data into the model, integrates temporal and spatial attributes to capture the spatial heterogeneity characteristics of the study object, and solves the spatiotemporal regression relationship with a local weighting method [43].
(15)$$\begin{array}{l} y_{i}=\beta_{0}(u_{i},v_{i},t_{i})+\sum_{k=1}^{d}\beta_{k}(u_{i},v_{i},t_{i})x_{ik}+\varepsilon_{i}\\ i=1,2,...,n \end{array}$$
where: (*x*_*i*1_, *x*_*i*2_,…, *x_id_*, *y_i_*) represents n groups of observed values of dependent variable y and independent variable *x*_1_, x_2_,…, *x_n_* at the observation point (*u_i_*, *v_i_*, *t_i_*); u_i_ and v_i_ are the longitude and latitude coordinates of the gravity center, respectively; (*u_i_*, *v_i_*, *t_i_*) is the space-time coordinate of the *i*th sample point; $\beta_{0}(u_{i},v_{i},t_{i})$ is the regression constant of point i; $\beta_{k}(u_{i},v_{i},t_{i})$ is the kth regression parameter of point i; *X_ik_* is the value of independent variable *x_k_* at point i. that is the value of each quantitative index in the index system of GTWR; $\varepsilon_{i}$ is the residual item of the model.

## 3. Results

### 3.1. Verification of Fitting Accuracy

When using the dynamic threshold method to demarcate the boundary between urban and rural areas, in order to ensure the best threshold is selected, the relative errors are counted. According to the relative errors, the extracted urban built-up area(S(DN)) is the closest to the actual built-up area (*S_stat_*). The relative errors extracted by each province from 2001 to 2019 are shown in Table 3, which are kept at −0.59~1.69%, with a low level. The results can be used to build CE fitting model.

In order to verify the accuracy of the estimated CE, the CE was calculated according to the energy consumption, and the CE published on the CEADs was selected for verification. The relative error between the estimated CE and the CE calculated according to energy consumption is −0.015–0.005% (Figure 3a). The fitting R^2^ between the estimated data and the CE published on CEADs is 0.71 (Figure 3b), which shows that simulated data can be further used to explore the spatial distribution pattern of CE and the heterogeneity of driving factors.

### 3.2. Spatial-Temporal Variation Characteristics of CE

#### 3.2.1. Time-Varying Characteristics 

The proportion of CE in the northwest remained unchanged in general from 2001 to 2019 (Figure 4), with Shaanxi Province leading the five provinces with an average annual contribution of 30.8%, followed by Xinjiang Province and Gansu Province with an average annual contribution of 28.4% and 19.5%, respectively. Due to its resource endowment, Qinghai Province contributes less than 10% of the annual average CE. Ningxia is rich in coal resources. Although the proportion is small, the proportion of CE shows a growing trend, and in 2019, it overtook Gansu Province.

According to the temporal trend of CE of Northwest provinces from 2001 to 2019 (Figure 4), the average CE of counties in northwest provinces shows a trend of increasing, and the growth rate is slowing down year by year, which has not yet reached its peak. Take 2015 as the node. The average CE of counties have a rapid growth trend before 2015. The average CE of each county increased from 0.52 Mt in 2001 to1.83 Mt in 2015. After 2015, the growth rate of CE slows down, which is lower than the growth rate in any previous period. It is due to China’s participation in the Paris Climate Conference and the active implementation of various emission reduction policies. In response to the One Belt and One Road initiative, all provinces actively promote industrial cooperation and improve energy technology levels. The adjustment of industrial structure and the improvement in energy utilization rate are enhanced through the communication of capital and technology among regions. In addition, the variance of CE among counties increased from 0.96 in 2001 to 2.77 in 2019, and the variance in 2019 is 2.89 times that in 2001. The variance of CE among counties shows an upward trend. The unbalanced distribution of transportation, resources, and population among counties in northwest China leads to great differences in energy consumption.

#### 3.2.2. Spatial Evolution Characteristics

The spatial pattern of CE in the northwest counties is divided by the second gradient line and the Qinling Mountains (Figure 5), showing a spatial distribution pattern of “high in the north and low in the south”, with the expansion direction mainly showing the trend of circle expansion centered on the high-value area of the provincial capital gradually spreading to the surrounding area and the axial expansion with the major traffic routes as the axis. The narrow topography of Gansu Province makes CE expand axially along the Hexi Corridor.

The overall CE in the northwest was low in 2001. From 2001 to 2005, CE in the high CE counties continued to rise without an obvious spreading to the surrounding areas. Most of the counties with CE above 1 million tons rely on local resource endowments for development. It shows an independent and fragmented distribution pattern, including Xincheng, Shaibak, and Tianshan in Xinjiang; Yumen, Jiayuguan, Baiyin, Xigu, Qilihe, and Chengguan in Gansu Province; Dawukou, Xixia, Jinfeng, and Xingqing in Ningxia; Yuyang and City Six District in Xi’an in Shaanxi Province; Xining and Haixi in Qinghai Province. From 2005 to 2010, the area of high-value CE areas expanded. In 2015, the CE was obviously diffused. It was obvious in Urumqi, Lanzhou, the border between Shaanxi and Ningxia. In 2019, CE continued to increase. Xinjiang, northern Shaanxi, and northern Ningxia continue to expand and form a dense CE region. In general, the spatial distribution is characterized by fragment aggregation. According to the statistics of the accumulative proportion of the number of counties and the corresponding CE, it is found that the spatial distribution pattern conforms to the Pareto principle.

### 3.3. Spatial Distribution Pattern

#### 3.3.1. Spatial Clustering Characteristics

The global Moran’s I is greater than 0 (Table 4), which demonstrates that the spatial pattern of CE in the northwest has obvious clustering distribution characteristics. Moran’s I index increased from 0.392 in 2001 to 0.505 in 2019. The spatial correlation degree of CE tends to be strengthened. The county with similar CE tends to be concentrated in space.

According to the Lisa clustering diagram of CE from 2001 to 2019 (Figure 6), the high-high aggregation areas are mainly distributed in the Karamay and Urumqi, the main city of Xi’an in Shaanxi, and the border between Ningxia and Shaanxi. The high-high agglomeration area has been expanding over time. The low-low agglomeration area has a stable distribution range, mainly concentrated in the southwest of Qinghai and the Qinling-Daba Mountains in Shaanxi Province, which is an ecological area. CE is strongly bound by the environment. Additionally, the central city itself is not developed enough to strongly drive the development of the surrounding cities, resulting in low-low agglomeration areas. 

#### 3.3.2. Spatial Heterogeneity Characteristics

The overall difference is divided into the difference between provinces and the difference within provinces. Moreover, analyze the contribution value of different provinces. The Theil index based on per capita CE ranges from 0.195 to 0.332 (Table 5). The per capita CE shows obvious regional differences, but the overall difference decreases. The within-province difference is larger than the between-province difference; its proportion decreases year by year. Thus, the overall difference in the Northwest region is caused by the within-province difference. The contribution rate of each province to the overall difference is: Shaanxi (27.58%) > Xinjiang (24.44%) > Gansu (17.88%) > Ningxia (11.56%) > Qinghai (6.71%). The contribution rate of Gansu Province to the overall variation has an obvious decreasing trend, while the other four provinces remain basically unchanged.

Compared with the regional differences in per capita CE, the regional differences in CE intensity are significantly smaller (0.077–0.122). The regional differences in CE intensity show a “V” curve from 2001 to 2019, which starts to rebound to the original value after falling to the lowest value in 2010. It is related to the economic recession caused by the global financial crisis in 2008, after which the economy in northwest China gradually recovered. The within-provincial variation based on CE intensity is also much higher than the between-provincial. The contribution of all five provinces to the overall variation shows a decreasing trend.

The spatial differences based on per capita CE and carbon intensity both show a “convergence within groups and divergence between groups”. In general, the Theil index based on per capita CE is much higher than the Theil index based on carbon intensity, which indicates that the matching degree between CE and GDP is higher than that between CE and population size.

The two-stage nested Theil index divides the study area into four levels of “region–province–city–county” to study the variability at each level. Results can be seen (Figure 7): (1) The overall difference in per capita CE has been decreasing, from 0.332 in 2001 to 0.195 in 2019. The overall difference in CE intensity has shown a “V” curve, which dropped to the lowest point in 2010 and then regressed. (2) Based on per capita CE, BS is the most important source of the overall variation. The average contribution rate was 47.44%. However, its contribution rate is decreasing. BC makes the most contribution to the spatial variation based on CE intensity. (3) Whether spatial differences are based on per capita CE or CE intensity, BP shows an increasing trend.

### 3.4. The Influencing Factors of CE

#### 3.4.1. Selection of Indicators

Considering the continuity and availability of multi-year data at the county scale, we selected six factors: population size (POP), industrial structure (IS), urbanization level (UR), land use scale (LUS), economic scale (GDP), and CE intensity (CEI). The resident population represents POP, which has a more continuous and stable frequency and intensity in the region. Population growth contributes to the rise of energy demand. Human activities promote urbanization, leading to a change in land use type, resulting in an increase in CO_2_ [44]. Fossil energy consumption caused by industrial development is the main source of CE [45]. The structural share of secondary GDP is chosen to represent the IS. CEI is the CE per unit of GDP. It can characterize the effect of technological progress on CE [36]. The impact of urbanization on CE follows the Environmental Kuznets curve. According to Wang’s study [46], the urbanization level of underdeveloped areas greatly impacts CE. Thus, the urbanization level is an important indicator [47]. In addition, some scholars believe that the urbanization of LUS is also a major factor affecting CE [38]. Urban land is the main area for human production and living and is the main carrier of a large number of infrastructures. The urban land expansion changes carbon sequestration capacity, resulting in increased CE. Economic development drives the input of production factors and the expansion of the production scale, which causes more energy consumption and CE [48]. GDP is the total output value of all industries in a period, so the study chooses GDP to characterize the economic scale.

#### 3.4.2. Temporal Trends of Influencing Factors

Considering the requirements of GTWR on multicollinearity, this paper checks the collinearity of influencing factors. The VIF (variance expansion factor) is less than 10, and there is no multicollinearity among the influencing factors. The R^2^ and the adjusted R^2^ are 0.798.

Figure 8 shows that the intensity of POP has been increasing yearly, with a continuous promoting effect, which presents that although the population of northwest China continues to rise, the population structure and consumption habits have not significantly improved. The intensity of IS is a coexistence of promotion and inhibition. The intensity increased from 2000 to 2010 and contracted after 2010. The overall effect is still promotion. This may be related to the fact that Jiuquan was approved to build China’s first 10-megawatt wind power base in 2008, and the northwest region has gradually stepped into the fast lane of new energy development. The rise of new energy industries has changed the fossil energy structure of IS and led the intensity of IS to shrink. 

The trend of UR is in line with the EKC principle. The overall urbanization level is generally low in northwest China. Moreover, the response of CE to urbanization is significant. The inhibition intensity of LUS increased year by year, which is related to the phenomenon that the expansion rate of urban land is higher than the inflow rate of population in China, and it also shows that there are unreasonable phenomena of the urban land expansion pattern and planning. The range of GDP action intensity gradually shrinks but still shows the promotion effect. The economic development is oriented toward the pursuit of the total growth in northwest China. Rough production mode, high input, and high energy consumption input-output processes have formed certain CE. However, it can be seen that the intensity of GDP decreases, indicating that the counties in northwest China gradually practice the development concept of a low-carbon economy. The intensity of CEI increases obviously. A low energy utilization rate makes economic development at the expense of more fossil fuel consumption.

#### 3.4.3. Spatial Distribution Characteristics of Influencing Factors

According to Figure 9, the effect intensity of POP on CE increases steadily. The average natural population growth rate was 8.86‰ in Xinjiang, ranking second in the national average population growth rate, with stable population growth. In addition, Xinjiang is also the province with the largest population inflow in the northwest region. Under the opportunity of The development of the western region and One Belt and One Road in Xinjiang, better work benefits and preferential policies attract the population flow to Xinjiang. Shaanxi, as the province with the best economic development in the northwest, the flow of talent to Shaanxi is an inevitable trend. Compared to the other four provinces, IS plays a prominent role in Xinjiang. Its huge energy reserves and long history of exploitation have made the proportion of secondary production in Xinjiang remain high since 2001. Shaanxi and southwestern Gansu have a higher level of urbanization. UR has a more obvious role in promoting CE. Qinghai and southwestern Xinjiang are subject to ecological constraints and have a slower pace of urbanization development, which has a slightly inhibitory effect on CE.

LUS has an inhibiting effect on CE. Its spatial distribution is wide. The Loess Plateau is located in northern Shaanxi, with the Qinling Mountains in the south. The continuous Qilian Mountains run through Gansu. The expansion area of urban development in the two provinces is limited. Xinjiang has a flat and vast territory. With the support of national policies, urban expansion is rapid. The spatial intensity of GDP is “strong in the center and weak in the east and west”. This indicates that the economic development of Shaanxi started early. It has entered the stage of low-carbon development, with stable growth in total volume and green and low-carbon structure, so the role of GDP in promoting CE is gradually decreasing. The effect of the economic growth model, which relies on energy consumption in Xinjiang, is also decreasing yearly. Due to the late start of economic development and unbalanced development of industrial structure, Xinjiang still lags behind in the development of high-tech and high-added-value products and lacks competitiveness. The counties along the Lanzhou-Xinjiang Railway are guided by national strategies. It is economically stimulated by many major regional infrastructures while also causing CE. After 2010, the emerging high-value area of CE at the junction of Mongolia, Shaanxi, and Ningxia has been expanding. Its energy utilization rate is low. The promotion effect of CEI on CE is more obvious.

## 4. Discussion

### 4.1. Discussion on Spatial Heterogeneity of Carbon Emissions in Northwest China 

From 2001 to 2014, the county carbon emissions in northwest China increased year by year, and the growth rate slowed down after 2015. This conclusion is consistent with the research of Dong et al. [49]. In 2000, China formally put forward the strategy of western development. With favorable national policies and strong financial support, the economic development of infrastructure construction in northwest China achieved remarkable results [50]. At that time, the industry in northwest China was still a resource-based industry, which focused on the production and processing of primary products. The rapid economic development was accompanied by huge environmental pollution [51]. With the arrival of the post-Kyoto era, developing countries have been included in the global carbon emission reduction responsibility list. In addition, China put forward the “double carbon” target in 2015, and the growth rate of carbon emissions in northwest China slowed down under the background of national emission reduction.

The differences between provinces in northwest China are gradually increasing, while the differences between cities and counties within each province are shrinking. The provincial governments actively use administrative power to coordinate the development of the provincial economy and have achieved certain results. From the perspective of the whole northwest region, the coordinated development effect among provinces is weak, and the urban agglomeration has not effectively exerted its driving effect [52]. In addition, the development among provinces has a competitive effect, which will hinder the effective circulation of resources and technologies between neighboring provinces, resulting in the development of each province being dependent on its own resource endowment, and the differences among provinces are increasing year by year.

The extensive economic development mode relying on energy consumption is still the main development mode in northwest China. Zhang et al. also think that the economic scale has the strongest positive driving effect on carbon emissions in northwest provinces [51]. After the approval of the Jiuquan wind power base construction in 2008, the new energy in the northwest region began to develop rapidly. The development of new energy has accelerated the transformation of the energy industry structure. However, the transformation of industrial structure cannot promote the improvement in the energy utilization rate, which depends on the continuous promotion of the technical level. At present, the technical level is not highly developed and has obvious differences in northwest China. Since 2001, the energy utilization rate has been the second driving factor after the economic effect, which is consistent with the conclusion of Su et al. [14]. 

### 4.2. Policy Implications

The difference between cities and counties is the main source of carbon emission difference in the northwest. The difference between provinces is increasing year by year. The focus of carbon governance lies in coordinating neighboring provinces, strengthening the mutual flow of talents, resources, and technologies among provinces, especially in the introduction and exchange of new energy technologies, breaking the barriers between provinces, promoting the balanced development among districts and counties within each province through policy guidance, and solving the problem of unbalanced and inadequate development of districts and counties. 

Carbon emissions in northwest China are obviously dependent on resources. The coal, energy, and chemical industry base has always been a high-carbon-emission area in northwest China. The realization of low-carbon development should break through the economic development path dominated by the traditional energy industry and set long-term phased goals for the energy structure dominated by coal. Northwest China should vigorously promote the energy structure with new energy as the main body. With the help of China’s vast market advantages and the larger market and regional investment level provided by the new “belt and road initiative” strategy, it is beneficial to reduce the cost of introducing and popularizing emission reduction technologies and improve energy utilization efficiency by upgrading the technical level.

The northwest region should take advantage of the development opportunities of the “One Belt and One Road” to vigorously develop the high-end equipment manufacturing industry, improving product quality and adjusting the industrial structure, especially the manufacturing industry related to rail transit, infrastructure construction, modern technology, and equipment transformation of traditional industries. The equipment manufacturing, energy, and transportation industries are the key industries under the construction of the “One Belt and One Road”. These industries inevitably generate a large amount of carbon emissions. Therefore, it is necessary to strictly limit the CE of high emission, high energy consumption, and surplus industries, strengthen energy conservation management of energy mining and processing industries, deeply develop carbon removal technologies, improve production efficiency and reduce carbon emissions through technological innovation.

## 5. Conclusions

This paper estimates CE in the counties in Northwestern China from 2001 to 2019 based on nighttime lighting data and population data, reveals the spatial-temporal pattern of CE in the northwest, and uses socioeconomic data to explore the heterogeneity of the influencing factors. The conclusions are as follows: The CE in northwest China shows a trend of the increasing total, but the growth rate is decreasing from 2001 to 2019. The carbon emission in northwest China has changed from rapid growth to steady growth. The spatial pattern forms a circle expansion trend with the high-value area of provincial capital and the resource-intensive area at the border between Inner Mongolia, Shaanxi, and Ningxia as the core and an axial expansion trend along the Hexi Corridor. The distribution of high-high aggregation areas is closely related to the resource endowment of counties. Constrained by the ecological environment, the Qinling-DaBa Mountains and southwest Qinghai are stable low-low agglomeration areas.

The spatial-temporal variability based on per capita CE and CE intensity are both notable. The between-provincial difference is smaller than the within-provincial difference. The result of the two-stage nested Theil index indicates that the contribution rate of BP is significantly lower than that of BS and BC, but the contribution rate of BP increases year by year. The overall variation shows a “convergence within groups and divergence between groups”. CE is influenced by economic scale to the greatest extent, followed by population scale and urbanization level. The intensity of the economic scale shrinks gradually. The main factors affecting CE have obvious spatial and temporal divergence and show different intensity characteristics in different regions.

This paper selects northwest China for research, which has received less attention and uses Landscan data and nighttime light data to spatialize provincial CO_2_ reasonably. The application of a two-stage nested Theil index provides a new research idea for the study of heterogeneity. The Geographically and temporally weighted regression intuitively displayed the heterogeneity of driving factors in each research individual. Limited to the availability and consistency of socioeconomic data, the driving factors selected in this article do not include all the influencing factors. Policy factors and the impact intensity of emergencies need to be further discussed. Carbon emission research is a very complex process involving different industries and fields. The carbon emission estimation results based on the statistical yearbook are only a relatively accurate value. Carbon in nature is always in dynamic change, and the interaction between carbon source and carbon sink is a complex and continuous dynamic process. The dynamic monitoring of the carbon change process in a large area with remote sensing is the focus and difficulty of future research.

## Figures and Tables

**Figure 1 ijerph-19-13405-f001:**
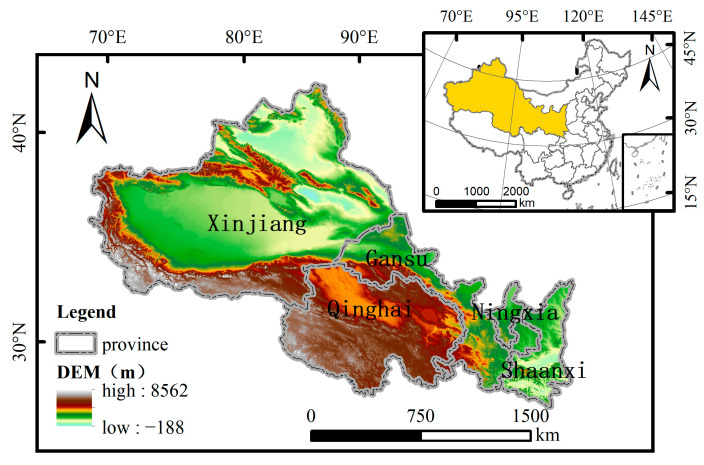
Topographic maps of the study area.

**Figure 2 ijerph-19-13405-f002:**
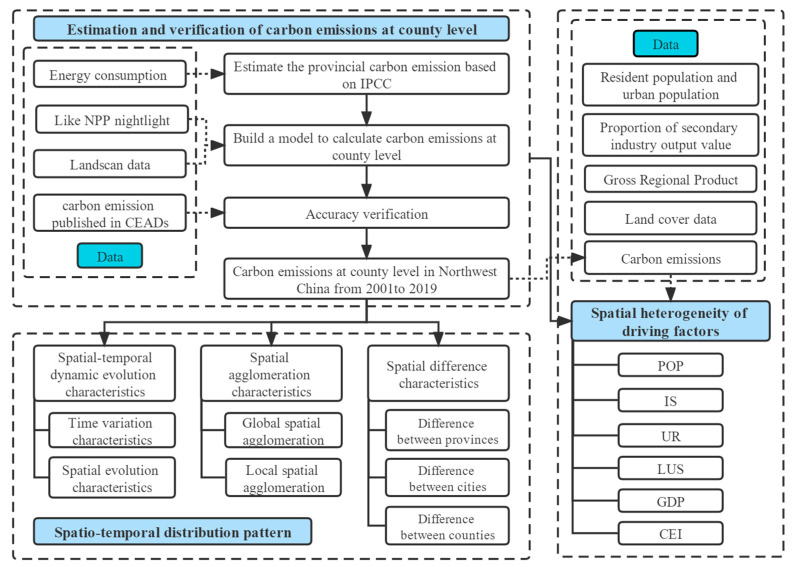
Flow-process diagram.

**Figure 3 ijerph-19-13405-f003:**
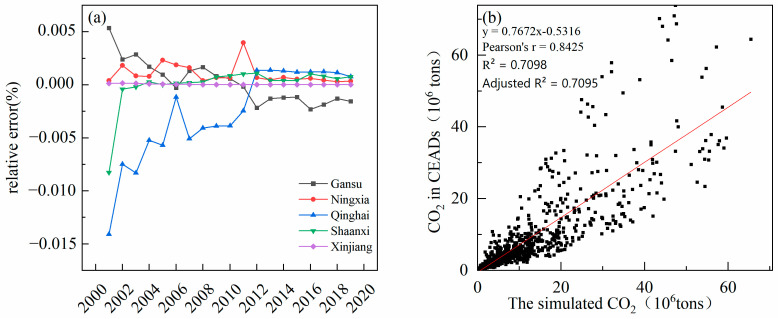
(**a**) Relative error between stimulated CE and CE calculated based on the statistical yearbook. (**b**) Fitting accuracy between stimulated CE and CE published on CEADs.

**Figure 4 ijerph-19-13405-f004:**
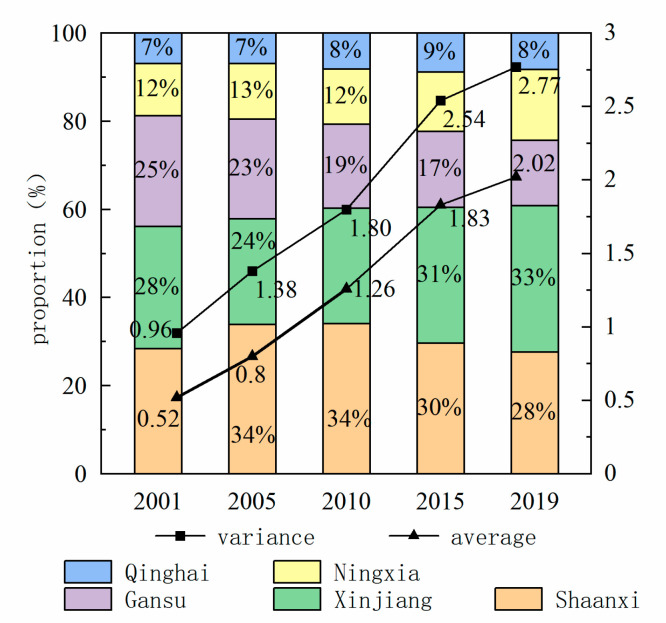
Time-varying trend of CE in northwest China at the county level from 2001 to 2019.

**Figure 5 ijerph-19-13405-f005:**
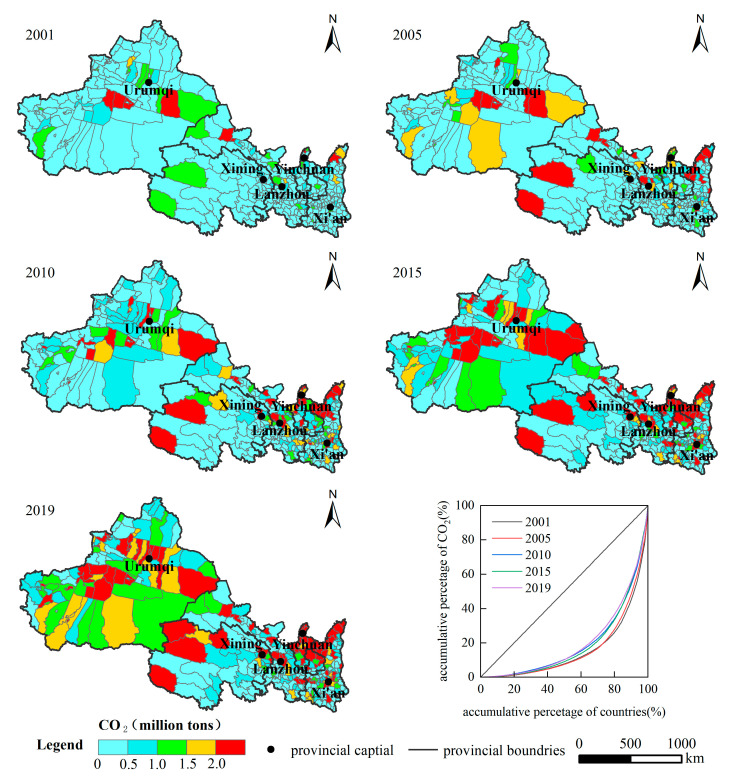
Spatial distribution of CE in northwest China at the country level from 2001 to 2019.

**Figure 6 ijerph-19-13405-f006:**
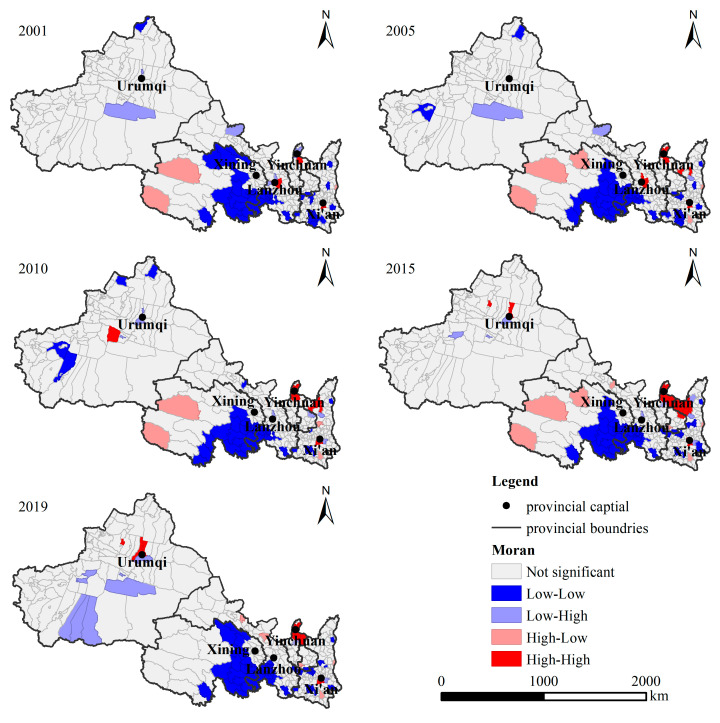
Spatial pattern of LISA of CE in northwest China at the country level from 2001 to 2019.

**Figure 7 ijerph-19-13405-f007:**
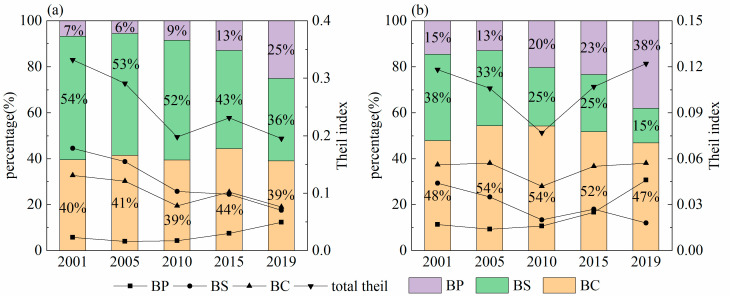
(**a**) Two-stage nested Theil index based on per capita CE. (**b**) Two-stage nested Theil index based on carbon intensity.

**Figure 8 ijerph-19-13405-f008:**
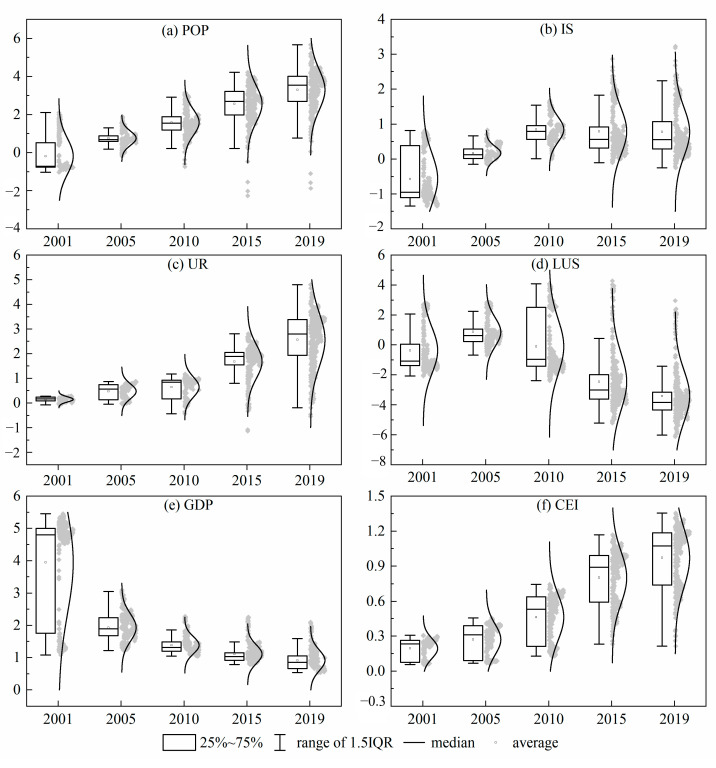
Time trend of the regression coefficients of factors.

**Figure 9 ijerph-19-13405-f009:**
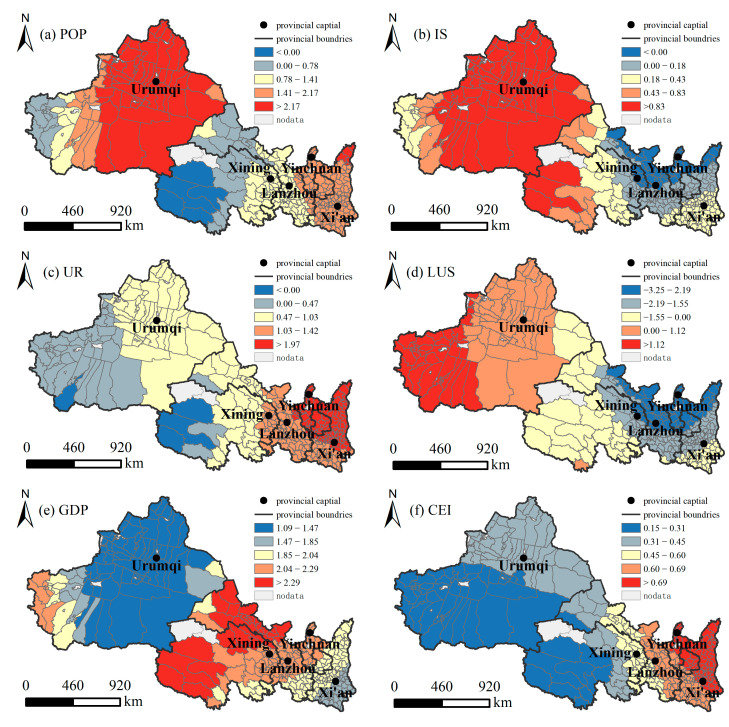
Spatial pattern of the mean of the regression coefficients of factors.

**Table 1 ijerph-19-13405-t001:** Data list.

Data	Spatial Resolution	Description	Data Source
Energy consumption		Includes 12 energies: raw coal, washed coal, other washed coal, coke, crude oil, gasoline, kerosene, diesel, fuel oil, natural gas, electricity, liquefied petroleum gas.	https://data.cnki.net/ (accessed on 11 March 2022)
Carbon emission		The data from 1997 to 2017 are published on the CEADs website at county level [40].	https://www.ceads.net.cn/ (accessed on 11 March 2022)
NPP-VIIRS-like data	500 m	The data are processed by an auto-encoder (AE) model, including convolutional neural networks. The R^2^ in urban and pixel scale is as high as 0.87 and 0.95, respectively [41].	https://doi.org/10.7910/DVN/YGIVCD (accessed on 15 March 2022)
Landscan data	About 1000 m	The study resampled the data to 500 m.	https://landscan.ornl.gov (accessed on 15 March 2022)
MCD12Q1	500 m	Used to construct LUS factor.	https://search.earthdata.nasa.gov/ (accessed on 18 March 2022)
Socioeconomic factor		Includes 4 factors: population, urban population, GDP, proportion of secondary industry output value.	https://data.cnki.net/ (accessed on 22 March 2022)

**Table 2 ijerph-19-13405-t002:** The carbon emission coefficient of various energies.

Fossil Energy	Coal Conversion Factors	Carbon Emission Coefficient
	(t Standard Coal/t)	(t Carbon/t Standard Coal)
Raw coal	0.7143	0.7559
Washed coal	0.9000	0.7559
Other washed coal	0.2857	0.7559
Coke	0.9714	0.8550
Crude oil	1.4286	0.5857
Gasoline	1.4714	0.5538
Kerosene	1.4714	0.5714
Diesel	1.4571	0.5921
Fuel oil	1.4286	0.6185
Natural gas	1.3300	0.4483
Electricity	0.3450	0.2720
Liquefied petroleum gas	1.7143	0.5042

Note: Data are from China Energy Statistics Yearbook (https://data.cnki.net/ (accessed on 11 March 2022)).

**Table 3 ijerph-19-13405-t003:** The relative error between extracted urban built-up area and the actual built-up area.

		Shaanxi	Gansu	Qinghai	Ningxia	Xinjiang	Total
2001	*S_stat_* (km^2^)	466.87	418.28	97.89	159.45	510.82	1653.31
	S(DN) (km^2^)	468.75	418.5	98.25	156.75	509.75	1652.00
	relative error (%)	−0.40	−0.05	−0.37	1.69	0.21	0.08
2005	*S_stat_* (km^2^)	561.65	507.44	105.92	248.94	595.50	2019.45
	S(DN) (km^2^)	559.75	506.25	105.75	248.25	594.00	2014.00
	relative error (%)	0.34	0.23	0.16	0.28	0.25	0.27
2010	*S_stat_* (km^2^)	758.48	632.80	113.88	343.79	838.21	2687.16
	S(DN) (km^2^)	759.75	635.00	112.50	341.50	840.25	2689.00
	relative error (%)	−0.18	−0.35	1.21	0.67	−0.24	−0.07
2015	*S_stat_* (km^2^)	1073.40	834.40	194.30	455.10	1185.40	3742.60
	S(DN) (km^2^)	1070.50	830.75	194.25	456.75	1185.50	3737.75
	relative error (%)	0.27	0.44	0.03	−0.36	−0.01	0.13
2019	*S_stat_* (km^2^)	1357.5	875.72	215.21	489.1	1421.61	4359.14
	S(DN) (km^2^)	1356.5	875.53	215.02	492	1424	4363.05
	relative error (%)	0.07	0.02	0.09	−0.59	−0.17	−0.09

**Table 4 ijerph-19-13405-t004:** Statistical table of global Moran’s I from 2001 to 2019.

	2001	2005	2010	2015	2019
Moran’s I	0.392	0.401	0.445	0.502	0.505
Z	12.855	13.193	14.761	16.636	16.675

**Table 5 ijerph-19-13405-t005:** Difference contribution rate of CE in northwest China from 2001 to 2019.

	Theil Index Based on per Capita CE					Theil Index Based on Carbon Intensity				
Year	2001	2005	2010	2015	2019	2001	2005	2010	2015	2019
Theil index	0.332	0.291	0.198	0.231	0.195	0.118	0.106	0.077	0.107	0.122
Gansu	23.683	21.570	17.668	15.159	11.331	24.371	24.068	21.119	19.993	16.486
Ningxia	10.563	12.005	11.377	11.740	12.128	5.234	5.397	5.420	5.012	4.224
Qinghai	6.389	6.393	7.291	7.558	5.918	4.516	4.894	4.485	4.650	3.691
Shaanxi	26.788	32.299	31.467	26.180	21.177	34.184	34.509	30.754	29.258	23.825
Xinjiang	25.753	22.163	23.676	26.358	24.228	17.096	18.026	17.795	17.663	13.805
Within province	93.177	94.429	91.480	86.996	74.782	85.400	86.894	79.574	76.576	62.032
Between province	6.823	5.571	8.520	13.004	25.218	14.600	13.106	20.426	23.424	37.968

## Data Availability

All data are available in the public domain.

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
