# Peer review of "Analysis of Dynamic Evolution and Spatial-Temporal Heterogeneity of Carbon Emissions at County Level along “The Belt and Road”—A Case Study of Northwest China"

_ijerph, 2022, doi:10.3390/ijerph192013405_

Round 1
Reviewer 1 Report
This paper was mainly analyzed the carbon emissions at county level in Northwest China, using both population data and nighttime light data. However, authors did not describe the algorithm, only sited a reference [31].
1. I suggest to describe the method simply in this manuscript. According to the website (https://doi.org/10.7910/DVN/YGIVCD), time series of nighttime light data only from 2000 to 2018.
2. Carbon emissions verification is suggested in the manuscript, compared with the CEADs (https://www.ceads.net.cn/).
Reviewer 2 Report
1. Title: Spatial pattern includes spatial heterogeneity and duplication should be avoided.
2. Abstract and Introduction: The research objectives of this paper should be clearly put forward, as well as the innovative contributions of this paper including new methods and new conclusions should be summarized.
3. The nighttime light remote sensing image data of relevant years need to be introduced in more detail, and it is suggested to add a table.
4. In the Discussion section, it is necessary to add the comparative analysis of the research results of this paper and other scholars.
5. In the Conclusion, it is suggested to supplement the deficiencies of this paper and the prospect of future research.
6. References:Chinese scholars have a large number of papers. It is suggested to increase the research literature of scholars from other countries.
Reviewer 3 Report
Dear Author(s),
The topic developed in your text is of interest. The research on greenhouse gas emissions and the evolution of territorial development –especially in developing countries– is a key issue in the sustainability context. This subject could provide an important contribution to this research area. However, in my opinion, your manuscript in its present form doesn’t support the sufficient quality for its publication in IJERPH. In my opinion, your text is not a “Research paper”, but rather it is a “Case study”. Accordingly, as research paper, is not suitable for publication in IJERPH. Publications in peer-reviewer journals –as IJERPH– are to disseminate knowledge. In this sense, there is still very significant room for improvement to be published. You can see, for this purpose, an interesting paper entitled, ‘How to write a paper for successful publication in an international peer-reviewed journal’ (Tress et al., 2014). You should review the “Guide for Authors” and/or consult with other colleagues to adjust your text to this type of paper. In any case, I think that your manuscript provides an interesting case study. Perhaps, it may be possible its publication in other Journals. I am completely sure of your ability to make progress on this subject. I encourage you to carry on down the path you have chosen.
Round 2
Reviewer 2 Report
1. There are still some spelling mistakes.
2. Table 2:Please add the data source.
Reviewer 3 Report
Dear Author(s),
Thank you very much for diligently considering the given remarks in your revision. I’ve recommended that your paper is suitable for publication.
